# The SOS Error-Prone DNA Polymerase V Mutasome and β-Sliding Clamp Acting in Concert on Undamaged DNA and during Translesion Synthesis

**DOI:** 10.3390/cells10051083

**Published:** 2021-05-01

**Authors:** Adhirath Sikand, Malgorzata Jaszczur, Linda B. Bloom, Roger Woodgate, Michael M. Cox, Myron F. Goodman

**Affiliations:** 1Department of Chemistry, University of Southern California, Los Angeles, CA 90089, USA; adhirats@usc.edu; 2Department of Biological Sciences, University of Southern California, Los Angeles, CA 90089, USA; jaszczur@usc.edu; 3Department of Biochemistry and Molecular Biology, University of Florida, Gainesville, FL 32611, USA; lbloom@ufl.edu; 4Laboratory of Genomic Integrity, National Institute of Child Health and Human Development, National Institutes of Health, Bethesda, MD 20814, USA; woodgate@nih.gov; 5Department of Biochemistry, University of Wisconsin-Madison, Madison, WI 53706, USA; cox@biochem.wisc.edu

**Keywords:** DNA damage-induced mutagenesis, *Escherichia coli* SOS regulon, DNA polymerase V mutasome, β processivity clamp, TLS

## Abstract

In the mid 1970s, Miroslav Radman and Evelyn Witkin proposed that *Escherichia coli* must encode a specialized error-prone DNA polymerase (pol) to account for the 100-fold increase in mutations accompanying induction of the SOS regulon. By the late 1980s, genetic studies showed that SOS mutagenesis required the presence of two “UV mutagenesis” genes, *umuC* and *umuD*, along with *recA*. Guided by the genetics, decades of biochemical studies have defined the predicted error-prone DNA polymerase as an activated complex of these three gene products, assembled as a mutasome, pol V Mut = UmuD’_2_C-RecA-ATP. Here, we explore the role of the β-sliding processivity clamp on the efficiency of pol V Mut-catalyzed DNA synthesis on undamaged DNA and during translesion DNA synthesis (TLS). Primer elongation efficiencies and TLS were strongly enhanced in the presence of β. The results suggest that β may have two stabilizing roles: its canonical role in tethering the pol at a primer-3’-terminus, and a possible second role in inhibiting pol V Mut’s ATPase to reduce the rate of mutasome-DNA dissociation. The identification of *umuC*, *umuD*, and *recA* homologs in numerous strains of pathogenic bacteria and plasmids will ensure the long and productive continuation of the genetic and biochemical journey initiated by Radman and Witkin.

## 1. Introduction

The inception of the SOS field occurred in 1953, a memorable year coincident with elucidation of the structure of DNA. The SOS concept emerged from Jean Weigle’s surprising observation that came to be known as Weigle reactivation. In brief, λ bacteriophage that were killed (inactivated) by exposure to UV radiation failed to lyse host *Escherichia coli*. However, infectivity was restored in the irradiated bacteriophage when the bacteria being infected were themselves also irradiated with ultraviolet light. There was, however, no free lunch, because *W*-reactivation was accompanied by *W*-mutagenesis, where the reactivated λ phage were heavily mutagenized [1]. In 1967, Evelyn Witkin proposed a far-reaching mechanism to explain *W*-reactivation, suggesting that a group of bacterial genes (now numbering > forty), were induced in response to chromosomal DNA damage [2]. Two regulatory genes played essential roles in the damage-inducible pathway: a repressor protein and a damage inducible protein that causes the inhibition of cell division (either directly, or indirectly). These two proteins were eventually identified as LexA and RecA [3,4,5,6].

Miroslav Radman, to whom this Cells volume and our article are dedicated, proposed a model to explain these observations, called SOS mutation-prone DNA repair. Although a formal manuscript was not published until 1974 [7], the real birth of the idea came four years earlier. In 1970, Miro circulated an informal written communication of the SOS hypothesis to his colleagues. Miro and Evelyn had both envisioned the existence of a new DNA polymerase induced in response to DNA damage, one which had exceedingly low fidelity. In 1977, a locus was identified, called *umuC.* When this locus was mutated, the cell became UV non-mutable, meaning that although mutations were present, they did not exceed spontaneous background levels [8,9]. In 1981, the *umuC* locus was shown to be a component of a DNA damage-induced operon regulated by LexA and RecA [10]. Cloning of the *umuC* locus revealed that it actually encodes two genes, which were named *umuD* and *umuC* [11,12]. Although induced as part of the SOS response, the UmuD protein is mutagenically inactive until it undergoes a RecA-facilitated autocatalytic cleavage reaction to generate mutagenically active UmuD’ [13,14,15]. UmuD’ can then interact with UmuC to form a UmuD’_2_C complex [16]. A soluble heterotrimer complex UmuD’_2_C was isolated and purified in 1996 [17] and was subsequently shown to copy undamaged primer-template (p/t) DNA and perform translesion DNA synthesis (TLS), but only if a RecA nucleoprotein filament (termed RecA*) was also present in the reaction [18]. Remarkably, however, TLS occurred in the absence of the replicative DNA polymerase III (pol III) [18]. In 1999, UmuD’_2_C was shown to be a DNA polymerase, with the pol active site located in the UmuC subunit [19,20].

On this occasion honoring Miro, it seems entirely appropriate to show Evelyn and Miro supping and likely also drinking together in Paris in 2012 (Figure 1A). There is yet another *French Connection* for our paper, Raymond Devoret (Figure 1B).

In 1989, Raymond identified a mutation in *recA* (*recA1730*, later identified as *recA* S117F) that was proficient, albeit somewhat compromised, in its ability to support homologous recombination and co-proteolytic cleavage of the LexA repressor and UmuD [21,22,23]. However, cells expressing RecA1730 were UV non-mutable [21], akin to *umuDC* mutants.

We subsequently found that these three proteins required for UV mutagenesis form an active mutasomal complex that copies undamaged and damaged DNA in the absence of RecA * [24]. There is a key role for RecA*, but not during DNA synthesis. Instead, RecA * is required to transfer a RecA monomer from its 3’-proximal tip to UmuD’_2_C to form UmuD’_2_C-RecA. However, UmuD’_2_C-RecA is inactive and requires a bound ATP (or the slowly hydrolysable form of ATP, ATPγS) in order to bind to p/t DNA [24,25], which is required to assemble the active form of pol V Mut = UmuD’_2_C-RecA-ATP [24,25]. In the presence of UV or chemicals that cause DNA damage, the induction of pol V Mut is accompanied by about a 100-fold increase in mutagenesis [26]. The increased mutations originate from misincorporations occurring directly opposite to DNA damaged bases, and also by mutations introduced when undamaged DNA is copied proximally upstream and downstream from a lesion [27,28].

Seemingly, the cell goes to great lengths to avoid suffering an even greater degree of mutagenesis by severely constraining pol V Mut’s presence in the cell so as to provide sufficient time for error-free DNA repair to occur prior to resorting to error-prone TLS. There are four levels of regulation imposed on pol V Mut to keep it in check: (1) temporal, where the levels of UmuD, UmuD’ and UmuC are tightly regulated by targeted proteolysis [29], so that the intracellular levels of UmuD’_2_C peak roughly 45 min post SOS induction [30]; (2) spatial, where UmuC is sequestered on the cell membrane in an insoluble form until it binds to UmuD’_2_ and enters the cytosol [31]; (3) internal, where pol V Mut has an intrinsic DNA-dependent ATPase that regulates the binding of the mutasome to p/t DNA [25]; (4) conformational [32,33], where pol V Mut exists in activated and deactivated forms depending on the location of RecA in relation to UmuC and UmuD’_2_. Two reviews describing the complex regulation of pol V Mut are contained in [34,35].

In this paper, we investigate how DNA synthesis, including TLS, is influenced by the β-sliding processivity clamp. Two forms of the mutasome were used: pol V Mut bound to wild-type RecA (pol V Mut WT) and pol V Mut bound to an SOS constitutive RecA mutant (pol V Mut E38K/∆C17). The constitutive mutasome is much more active and mutagenic compared to its wild-type counterpart and is responsible for causing ~100-fold increase in mutagenesis in the absence of chromosomal DNA damage [26]. A comparison of the biochemical properties of these two mutasomal forms provides insight into their distinctive in vivo behavior.

## 2. Materials and Methods

### 2.1. DNA Oligonucleotides

DNA oligonucleotides were purchased from Integrated DNA Technologies (Coralville, IA, USA) and their sequences are listed in Table 1. All oligomers were purified in-house using denaturing PAGE purification. The templates contained a 5’ phosphate modification in order to circularize the template. Briefly, the template to be circularized was annealed to a 20 mer linker, which was complementary to the first ten and the last ten nucleotides of the linear template. After annealing, we used T4 DNA ligase on the template/linker to connect the 5’-P and 3’-OH, forming a circle. The circle was then purified of its linear product and the linker by running through a PAGE gel and subsequent extraction.

### 2.2. Proteins

His-tagged pol V was purified from *E.coli* strain RW644, as previously described [36]. The His-tag on pol V does not affect the DNA synthesis function of the protein in any *in vitro* work [32,33,36]. RecA WT and RecA E38K/∆C17 were purified as previously described [37].

The β-clamp was expressed from a pET15 vector containing the *dnaN* gene and purified based upon previously described protocols [38], but with a few modifications. Briefly, the vector was transformed into BL21 (λDE3) cells and plated on LB media with agar and ampicillin. Individual colonies were picked from the plate and used to inoculate a 1 L culture. The cell pellet was lysed in a lysis buffer (20 mM Tris pH 7.5, 0.5 mM EDTA, 50 mM NaCl, 5 mM DTT and 10% glycerol) using a French press, three times. The cell lysate was subject to centrifugation and the supernatant saved for ammonium sulfate precipitation (70% saturation). The precipitated protein was dissolved and dialyzed in buffer (20 mM Tris-HCl pH 7.5, 0.5 mM EDTA, 5 mM DTT, 50 mM NaCl, and 10% glycerol). The protein was loaded onto two HiTrap Q columns and washed with a gradient rising from the initial buffer (20 mM Tris-HCl pH 7.5, 0.5 mM EDTA, 5 mM DTT, 50 mM NaCl, and 10% glycerol) to the final buffer (initial buffer with 1 M NaCl). Fractions containing the β-protein were pooled and dialyzed against buffer (10 mM sodium acetate pH 7.5, and 0.5 mM EDTA). The dialyzed protein was then loaded onto two HiTrap Heparin columns. At first, when the column was washed with buffer at pH 7.5, β did not bind to the column but contaminating proteins did. The eluted protein was brought to pH 6 and loaded onto a newly equilibrated column. The column was washed with a gradient from the initial buffer (10 mM sodium acetate pH 6.0, and 0.5 mM EDTA) to the final buffer (initial buffer with 1 M NaCl). Finally, the purest fractions were stored in buffer containing 20 mM Tris-HCl pH 7.5, 0.5 mM EDTA, 5 mM DTT, and 10% glycerol. Proteins were chromatographically pure and free of detectable endo- or exonuclease activities. The β-protein concentration was determined by measuring absorbance in denaturing conditions at 280 nm.

In order to purify the γ-complex clamp loader, the *holA* (δ), *holB* (δ’), *holC* (χ), *holD* (ψ), and short form *dnaX* (γ) were co-expressed from pET-Duet vectors to form the γ-complex clamp loader in *E. coli* BL21 (λDE3) cells and purified based on previously described protocols [39], but with a few modifications. Briefly, the pET-Duet vectors were transformed into BL21 (λDE3) cells and plated on LB media with agar and kanamycin and ampicillin. Individual colonies were picked from the plate and used to inoculate a 2 L culture. The cell pellet was lysed in a lysis buffer (20 mM Tris pH 7.5, 0.5 mM EDTA, 5 mM DTT, 50 mM NaCl, and 10% glycerol) using a French press, three times. Lysed cells were then subject to centrifugation and the lysate supernatant was loaded onto two HiTrap Heparin columns (GE Healthcare, Chicago, IL, USA). After loading the protein, the column was washed with a gradient running from the initial buffer (20 mM Tris-HCl pH 7.5, 0.5 mM EDTA, 5 mM DTT, and 10% glycerol) to a final buffer (initial buffer plus 1 M NaCl) with the γ-complex proteins eluting around 170 mM NaCl. The fractions containing protein were dialyzed into buffer (20 mM Tris-HCl pH 7.5, 0.5 mM EDTA, 100 mM NaCl, 5 mM DTT, and 10% glycerol) and loaded onto MonoQ column (Pharmacia Biotech, Uppsala, Sweden). After loading the protein complex, the column was washed with a gradient from the initial buffer (20 mM Tris-HCl pH 7.5, 0.5 mM EDTA, 100 mM NaCl, 5 mM DTT, and 10% glycerol) to the final buffer (initial buffer with 1 M NaCl). The γ-complex clamp loader eluted around 250 mM NaCl from the MonoQ column. Clamp loader complexes were stored in a buffer containing 20 mM Tris-HCl pH 7.5, 0.5 mM EDTA, 5 mM DTT, and 30% glycerol.

### 2.3. Pol V Mut Assembly

Pol V Mut was assembled following previously published protocol [25]. Briefly, RecA nucleoprotein filaments (RecA *) were assembled on Cyanogen Bromide Sepharose resin with covalently attached 45 nt ssDNA (3’-tip exposed). Any excess of unbound RecA and ATPγS was removed by extensive washes on small spin columns. His-tagged pol V was incubated with RecA * and a pol V Mut (UmuC-UmuD’_2_-RecA) complex was assembled. The concentration of pol V Mut was determined by SDS-PAGE, using purified pol V and RecA as protein concentration standards.

### 2.4. Activity Assays of Pol V Mut

The activity of pol V Mut wt, or pol V Mut E38K/∆C17, was measured at 37 °C and 30 °C on radioactive P^32^ (phosporus-32)-labeled 20 mer primer/template (25 nM; regular template, template V1, and Template V2 listed in Table 1) in the presence of ATP or ATPγS (500 μM) and dNTP substrates (mix of dTTP, dGTP, dCTP or mix of dATP, dTTP, dGTP, dCTP 500 μM each) in a standard reaction buffer containing 20 mM Tris (pH 7.5), 8 mM MgCl_2_, 5 mM DTT, 0.1 mM EDTA, 25 mM sodium glutamate and 4% (v/v) glycerol. For reactions with β-clamp and γ-complex, substrate DNA (25 nM) was incubated with a reaction buffer and ATP/ATPγS for one minute, followed by the addition of γ-clamp (200 nM), 8 times in excess to DNA and γ-complex (150 nM); 6 times in excess to DNA. Pol V Mut (400 nM) was added to the reaction and after one-minute incubation, the synthesis reaction was initiated by adding the dNTPs mixture. Aliquots were taken from the main reaction at appropriate time points and reactions were terminated with a stop solution containing 20 mM EDTA in 95% formamide. The p/t DNA product molecules were separated using 20% denaturing PAGE gel. Gel band intensities were measured by phosphorimaging with ImageQuant software, and the fraction of extended primer (% PE) was calculated by integrating the band intensities of extended primer DNA divided by the total integrated DNA band intensity. Each experiment was repeated 3 times, and the average % PE (percent p/t DNA extended) with the SD (standard deviation) was plotted for each reaction time point.

## 3. Results

A model sketch depicts individual steps in the regulation of pol V Mut (Figure 2). The first step is the conversion of pol V (UmuD’_2_C) to pol V Mut (UmuD’_2_C-RecA). This initial transactivation step requires the transfer of a molecule of RecA from the 3’-proximal-end of a RecA nucleoprotein filament (RecA *) to pol V to form an inactive mutasome (State 1). Assembly into an active mutasome, able to copy undamaged DNA and perform TLS, occurs when a molecule of ATP (or ATPγS) binds to form activated pol V Mut (UmuD’_2_C-RecA-ATP/ATPγS) (State 2). Binding of ATP triggers a conformational change needed for pol V Mut to bind to a primer/template 3’-OH end [33]. After TLS, pol V Mut can deactivate and is no longer able to synthesize DNA (State 3) [32,34]. This model was derived from biochemical studies with a “stand-alone” pol V Mut, purified free of RecA *, in which DNA synthesis occurred in the absence of the β-sliding processivity clamp [24,25,32].

### 3.1. Stimulation of Pol V Mut on Undamaged DNA

Here, we investigate pol V Mut-catalyzed DNA synthesis in the presence of β while copying undamaged primer/template (p/t) DNA (Figure 3). The experiments were performed using two forms of the mutasome, assembled either with wild-type RecA, pol V Mut WT, or with a constitutively induced form of RecA (RecA E38K/∆C17), pol V Mut E38K/∆C17. Loading and removal of the β-processivity clamp at a 3’-OH-p/t end occurs through the action of the γ-clamp loading complex (Figure 3A) [40,41,42]. A circular DNA template (90 nt) with 20 nt primer attached was used to prevent β from sliding off a primer-template (p/t) DNA end.

When present in an activated state, pol V Mut WT bound to ATP (pol V Mut WT-ATP) is barely able to synthesize DNA in the absence of β (Figure 3B) [25,32]. Although a primer extension band corresponding to the addition of 1 nt is clearly visible on the gel, along with faint bands for the addition of a few nucleotides, nevertheless the fraction of extended primers does not exceed 1%. A plausible explanation for the marginal primer extension is likely the short residence time of pol V Mut WT-ATP on p/t DNA (1.9 ± 0.5 s), which was measured at single-molecule resolution using TIRF-FRET [32]. More robust DNA synthesis was observed for pol V Mut WT-ATPγS with ~5% of the primers extended at 60 min (residence time = 5.6 ± 0.7 s, [32]). The addition of β strongly stimulates (~four-fold) pol V Mut WT-ATP (Figure 2B; ~4% primer extension occurs at 60 min, compared to ~1% in the absence of β). Pol V Mut WT-ATPγS shows relatively strong synthesis at 15 min in the absence of β, with bands extending more than half-way up the gel, while full-length 70 nt extension to the end of the 90 nt p/t was observed at 30 min (Figure 3B). The presence of β strongly stimulates pol V Mut WT-ATPγS, as the enzyme extends 10% of the primer at 15 min and more than 40% at 120 min (Figure 3B).

Compared to the wild-type mutasome containing ATP, the SOS constitutive pol V Mut E38K/ΔC17-ATP synthesized DNA to a far greater extent. Here, we observed 6% primer extension at 30 min and 10% at 120 min (residence time = 3.0 ± 0.4 s, [32]) (Figure 3C). Robust synthesis for pol V Mut E38K/ΔC17-ATP occurred in the presence of β, where 31% of the primers were extended at 15 min and 77% at 120 min (Figure 3C). Extension to the end of the 90 nt p/t occurred for reaction times ≥15 min. The presence of bands extending beyond the 70 nt template end were most likely caused by strand displacement on the circular p/t DNA (Figure 3C). In the absence of β, DNA synthesis is much greater with ATPγS compared to ATP, with full length synthesis occurring for reactions ≥15 min (Figure 3C).

An unanticipated result was observed when synthesis for pol V Mut E38K/ΔC17-ATPγS occurred in the presence of β (Figure 3C, ATPγS + β/γ lane), where pol V Mut E38K/ΔC17-ATPγS showed lower levels of activity with β-clamp (42% primer extension at 120 min, Figure 3C fourth panel) than pol V Mut E38K/ΔC17-ATP (77% primer extension at 120 min; Figure 3C second panel). Note that in the presence of ATPγS, γ-complex remains bound to β because ATP hydrolysis is required for active dissociation of γ [40]. Thus, the weaker stimulatory effect of ATPγS could result from competition between the UmuC subunit of pol V and the γ-clamp loader bound to β. It has been shown that pol V and γ-complex bind to β at the same or proximal binding sites, as is the case for each of the five *E. coli* pols [43,44].

Stimulation by β is roughly similar for pol V Mut WT-ATPγS and pol V Mut E38K/∆C17-ATPγS. However, the following question arises: why does β stimulate the activity of pol V Mut E38K/∆C17-ATP (77% extension at 120 min) to a higher extent than pol V Mut WT-ATP (5% extension at 120 min)? We suggest that since pol V Mut WT-ATP binds to p/t DNA with much lower affinity compared to pol V Mut E38K/∆C17-ATP [34], perhaps pol V Mut E38K/∆C17-ATP binds far more avidly to p/t DNA containing the β-clamp than does pol V Mut WT-ATP. A quantitative determination for the amount of primer extension (%PE) was obtained by integrating the gel band intensities for each lane and is plotted as a function of incubation time (Figure 3D,E).

### 3.2. β-Clamp Is Required for Pol V Mut-Catalyzed TLS

We used pol V Mut ± β to copy the circular DNA template containing an abasic lesion (X) (Figure 4). Since pol V strongly favors the incorporation of dA opposite X during transactivation with RecA * [45], we have excluded T from the template to allow us to measure pol V Mut-catalyzed TLS ± dATP. To exclude the possibility that dATP is used by the γ-complex to load β-clamp onto DNA [46], we first loaded β-clamp using γ-complex with an excess of ATP/ATPγS and then initiated the synthesis reaction with mix of dNTPs ± dATP.

With X located 3 nt from the primer-3’-end (Figure 4A–C), in the absence of dATP, pol V Mut WT was unable to perform TLS, nor could it incorporate directly opposite X with either ATP (pol V Mut WT-ATP) or ATPγS (pol V Mut WT-ATPγS), even with β/γ present in the reaction (Figure 4B). For pol V Mut WT, substantial TLS occurred in the presence of dATP when ATPγS and β/γ were included in the reaction (35% PE at 120 min), with processive extension of primers reaching the end of the template for t > 30 min (Figure 4C, right-hand gel). TLS did not occur with β/γ when ATPγS was replaced by ATP (Figure 4C, 2nd gel from the left). With X located 30 nt downstream from the primer terminus (Figure 4D), TLS was also observed as before, with dATP, ATPγS and β/γ (Figure 4E), although to a considerably smaller extent.

The constitutive mutasome, pol V Mut E38K/∆C17, is much more adept at TLS compared to pol V Mut WT, under a broader variety of conditions (Figure 5). 

When X is located 3 nt from the primer-end, with ATP (i.e., pol V Mut E38K/∆C17-ATP) in the absence of dATP, a small yet detectible level of incorporation of (most likely) G opposite X was observed, but TLS (i.e., extension beyond X) did not occur (Figure 5A, left-hand gel). The rate of dG incorporation opposite X is ~10-fold less than dA [45]. With β/γ and ATP included in the reaction, in the absence of dATP, TLS readily occurred, accompanied by full-length primer extension (Figure 5A, 2nd gel from left). A reduced stimulation by ATPγS occurs with β/γ (Figure 5A, two right-hand panels), in accord with data on p/t DNA without an abasic site (Figure 3C, two right-hand panels). In contrast, the inclusion of dATP resulted in a substantial increase in the rate of TLS for pol V Mut E38K/∆C17 containing either ATP or ATPγS in the presence of β/γ (Figure 5B). The TLS data are essentially the same with X located 30 nt from the p/t DNA end (Figure 5C). In summary, the key findings are that pol V Mut E38K/∆C17 requires the presence of β-clamp to perform TLS (Figure 5), and that both the ATP and ATPγS forms of the mutasome are proficient in copying past the lesion.

## 4. Discussion

This study extends our understanding of SOS mutagenesis (and the overall story initiated by Radman and Witkin) by defining the effects of the β-clamp on pol V Mut function. The β-processivity clamp facilitates DNA synthesis by both a wild-type and a constitutive mutasome on undamaged and damaged DNA, expanding upon earlier observations [32]. Previous studies have shown a stimulatory effect of the β-clamp on the processivity of pol V in the presence of RecA * [36,45,47,48,49]. Here, we report for the first time a strong stimulatory effect of β-clamp during synthesis on undamaged DNA and TLS, using the stand-alone active form of pol V, i.e., pol V Mut, in the absence of RecA *. The β-clamp strongly stimulates the fraction of p/t DNA elongated, pol processivity, and TLS for mutasomes assembled with either ATP or ATPγS cofactors (Figure 3 and Figure 4). The biochemical data showing more avid normal DNA synthesis and TLS with mutasomes containing ATPγS reflect the inability of intrinsic pol ATPase to hydrolyze ATPγS. The effects observed in the presence of ATP, where the pol V Mut is minimally active in the absence of β-clamp, have particular biological relevance. We have previously shown that hydrolysis of a single bound ATP is sufficient to release pol V Mut from p/t DNA [25].

Despite having three SOS induced DNA polymerases, pol II [50,51], pol IV [52] and pol V [19,20], in *E. coli*, the lion’s share of mutations that result from UV and chemical-induced chromosomal DNA damage is attributable to pol V-catalyzed TLS [8,9]. In wild-type cells, mutations attributable to pol V occur almost solely in response to DNA damage, although all three SOS pols contribute mutations, principally in deep stationary phase cells, that enhance cell fitness in the absence of external stressors [53,54]. In the absence of SOS induction, pol V is present at roughly five molecules/cell, as revealed by live-cell imaging using fluorescent-tagged pol V [31], which is below the level of detection by Western blotting using antibodies against UmuC [55]. Following SOS induction, pol V is present at ~50 molecules/cell accompanied by a roughly 100-fold increase in the level of mutagenesis [26]. 

This significant increase in mutagenesis reflects the action of pol V-catalyzed TLS, where mutations occur directly opposite a lesion, but extensive untargeted mutagenesis also occurs during gap-filling DNA synthesis by pol V on undamaged DNA, both upstream and downstream from the lesion [27]. In the absence of DNA damage, in contrast to wild-type *E. coli*, cells containing the constitutively expressed *recA* mutant (*recA730* [56] later identified as *recA*E38K [57]) exhibit ~100-fold increase in mutations caused by the induced pol V acting on undamaged DNA, whereas mutations are increased in a dose dependent manner in UV irradiated *recA*E38K cells [58]. The pol V mutasomes of course differ in the two *recA* genetic backgrounds; the inducible pol V is bound to a molecule of wild type RecA and the constitutive pol V is bound to a molecule of RecAE38K. Although both complexes must bind one molecule of ATP to form an active mutasome (Figure 2), and both are conformationally regulated in the same way [34], each mutasome has distinctive biochemical properties. Compared to wild-type pol V Mut, the constitutive mutasome exhibits increased activity and processivity, and remains physically bound to p/t DNA for a longer period of time [32].

The biologically relevant data are for when ATP (not ATPγS) is present in the activated mutasomal complex (Figure 2, State 2). When synthesis is carried out on undamaged DNA, in the absence of β, pol V Mut WT-ATP extends p/t DNA by at most one nt (Figure 3B). In contrast, pol V Mut E38K/∆C17-ATP is 5–10-fold more active over a 30 min to 2 h incubation time (Figure 3B,C). The increased activity for pol V Mut E38K/ΔC17-ATP is consistent with a ~2-fold longer residence time on p/t DNA compared to pol V Mut WT-ATP, determined by TIRF-FRET microscopy [34]. The activities of both mutasomes were stimulated substantially by β. DNA elongation rates were increased by ~5-fold for pol V Mut WT-ATP and ~7-fold for pol V Mut E38K/ΔC17-ATP (Figure 3B,C). In the presence of β, pol V Mut E38K/ΔC17-ATP copied the entire 70 nt ss template region and continued synthesis via primer strand displacement on the circular DNA template (Figure 3C, 2nd left-hand gel). A similar strong stimulatory effect of β was also seen during TLS by pol V Mut E38K/∆C17-ATP (Figure 5B,C).

This stabilizing effect of β suggests, perhaps, that along with its canonical role in tethering pol V Mut at 3’-primer termini, β might have a possible second role in inhibiting the intrinsic DNA-dependent ATPase of pol V Mut to reduce the rate of mutasome dissociation. A slight perturbation at the RecA-UmuD’_2_C interacting surface interface might be sufficient to reduce the efficiency of ATP hydrolysis. Inhibition of the ATPase would reduce the rate of pol V Mut-p/t DNA dissociation, which could provide enough time for pol V to perform gap-filling repair synthesis [27]. To address the question experimentally as to whether β affects the rate of DNA-dependent ATPase of pol V Mut, a new experimental set-up needs to be designed where we can distinguish between the weak rate of pol V Mut ATPase from the much stronger ATPase activity of the γ-complex used to load β onto DNA. This promises to be a challenging task but one that is certainly important to address in the future to determine if and how the intrinsic DNA-dependent ATPase of pol V Mut modulates polymerase activity.

How does pol V gain access to DNA *in vivo?* This has been an open question dating from the 1985 Bridges–Woodgate model in which TLS is mediated by the assembly of pol III, UmuC, UmuD, and RecA at a template damage site [59]. Although the model’s details have been revised over the years to include new biochemical data, for example, the replacement of UmuD by UmuD’ [60] and TLS performed by pol V (UmuD’_2_C) + RecA * in the absence of pol III [60], the basic tenet of the model has been retained: pol V-catalyzed TLS is *cis*-activated by RecA * assembled in a DNA gap proximal to a template lesion. However, the ability of RecA * to activate pol V when located in *trans* (Figure 2, State 1) [49], and the subsequent assembly of pol V Mut as a “stand-alone” polymerase that copies DNA in the absence of RecA * (Figure 2, State 2) [24] suggests that pol V mutasomes are able to diffuse freely within the cytosol to copy either damaged or undamaged DNA. Indeed, in cells containing *recA*E38K, mutations are made on undamaged DNA, implying that the ordered assembly of pol V Mut (Figure 2) is both necessary and sufficient to copy undamaged DNA and catalyze TLS, irrespective of where RecA * is located in the cell.

Launched more than 50 years ago by our creative colleague Miro Radman, SOS mutagenesis continues along an unabated voyage. Looking to the future, there are many examples of pathogens, cells and transmissible plasmids that encode UmuC, UmuD, and RecA homologs, providing a fresh venue for research, one having important implications for human health. Dedicating our SOS pol V mutasome *Cells* paper to Miro is a singular honor and pleasure and we conclude by wishing Evelyn Witkin (Figure 1A) a happy 100th birthday (9 March 2021), and remembering warmly Raymond Devoret (Figure 1B), and Harrison (Hatch) Echols, who coined the term mutasome [61,62] and, along with Bryn Bridges, served as mentors to Roger Woodgate.

## Figures and Tables

**Figure 1 cells-10-01083-f001:**
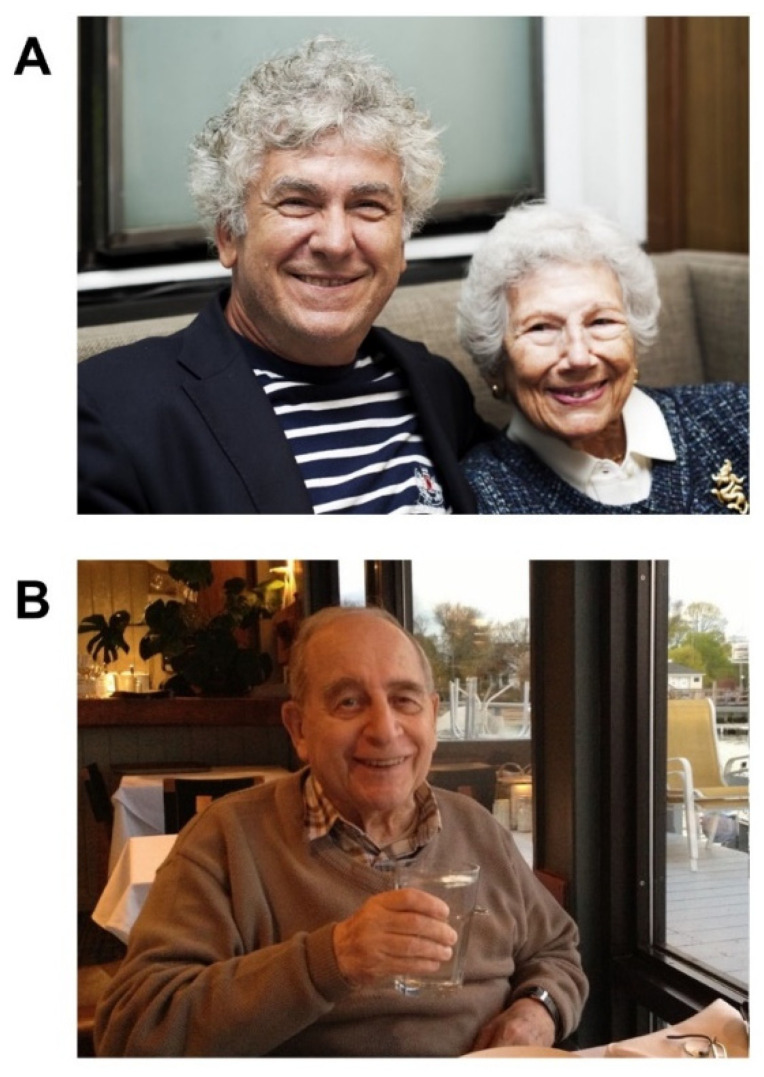
(**A**). Miro Radman (left) with Evelyn Witkin (right) in Paris, 2012. (**B**). Raymond Devoret à pris dîner à Sage Grill & Oyster Bar, New Haven, CT, 2013.

**Figure 2 cells-10-01083-f002:**
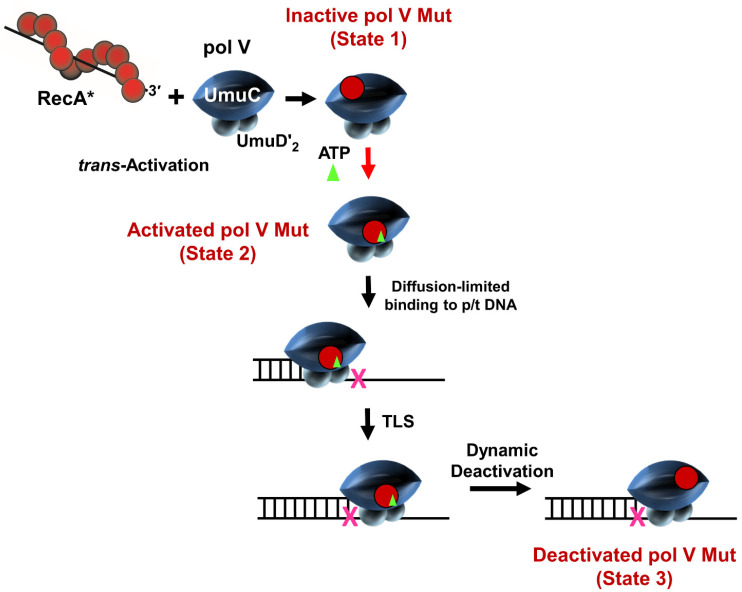
Regulation of pol V Mut. The first step in regulation of pol V Mut is the transfer of a single RecA monomer from a 3’ end of a RecA nucleoprotein filament (RecA *) to form pol V Mut (UmuD’_2_C-RecA). This protein complex is in an inactive state (State 1) and cannot bind to p/t DNA. A subsequent addition of ATP/ATPγS activates pol V Mut (active state = UmuD’_2_C-RecA-ATP; State 2). Due to ATP induced reorientation of RecA relative to UmuC and UmuD’ molecules, state 2 can bind to p/t DNA and synthesize DNA, including TLS (X denotes a DNA template lesion). After performing one cycle of DNA synthesis, the pol V Mut deactivates dynamically (State 3) and cannot perform any more rounds of DNA synthesis.

**Figure 3 cells-10-01083-f003:**
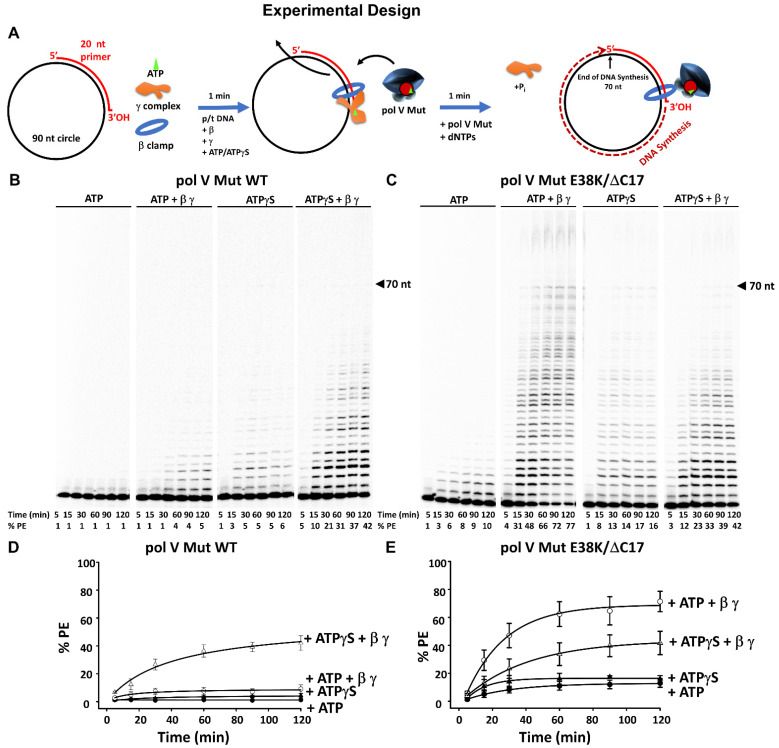
Effect of β-processivity clamp on activity of Pol V Mut. (**A**) Sketch showing experimental set up. Briefly, β-clamp is loaded on circular p/t DNA with γ-complex and ATP/ATPγS. Activity of pol V Mut WT (400 nM) (**B**), or pol V Mut E38K/∆C17 (400 nM) (**C**), was measured with 20 nt primer/ 90 nt circle (25 nM) in the presence of saturating concentration of ATP or ATPγS (500 μM) and dNTP’s (mix of dTTP, dCTP, dGTP 500 μM each). β-clamp and γ-complex are present in excess of DNA at 200 nM and 150 nM concentration, respectively. The experiments were repeated 3 times and the average % PE (percent p/t DNA extended) with the SD (standard deviation) for each reaction time point is graphed in panel (**D**) for pol V Mut WT and (**E**) for pol V Mut E38K/∆C17.

**Figure 4 cells-10-01083-f004:**
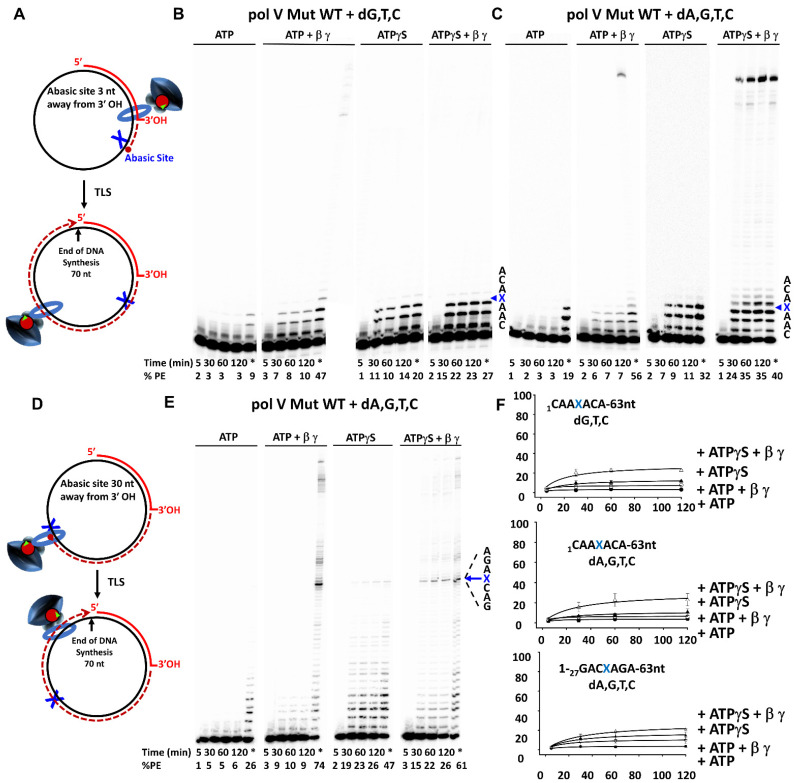
Translesion synthesis (TLS) by Pol V Mut Wild-Type (WT) on p/t DNA containing abasic site. (**A**,**D**) Sketch showing TLS by pol V Mut wt in presence of β-clamp on p/t DNA containing abasic site located 3 nt (**A**) and 30 nt (**D**) from 3’-OH junction. (**B**,**C**,**E**) TLS of pol V Mut WT (400 nM) was measured on circular p/t DNA (20 nt primer/ 90 nt circle; 25 nM) in the presence and absence of β-clamp and γ-complex with saturating concentration of ATP/ATPγS (500 µM) and with two sets of dNTPs: (B) dGTP, dTTP, dCTP and (**C**,**E**) dATP dTTP, dCTP, dGTP (500 µM each). (**B**,C**)** activity of pol V Mut WT on p/t containing an abasic site 3 nucleotides away from 3’-OH of primer. (**E**) activity of pol V Mut WT on p/t containing abasic site 30 nucleotides away from 3’OH of primer. The abasic site is marked by X and nucleotides surrounding the abasic site highlighted in the figure. At 120 min, 400 nM of RecA * was added to the reaction to trans-activate the mutasome which is represented by *. The experiments were repeated 3 times and the average % PE (percent p/t DNA extended) with the SD (standard deviation) for each reaction time point is graphed in panel (**F**).

**Figure 5 cells-10-01083-f005:**
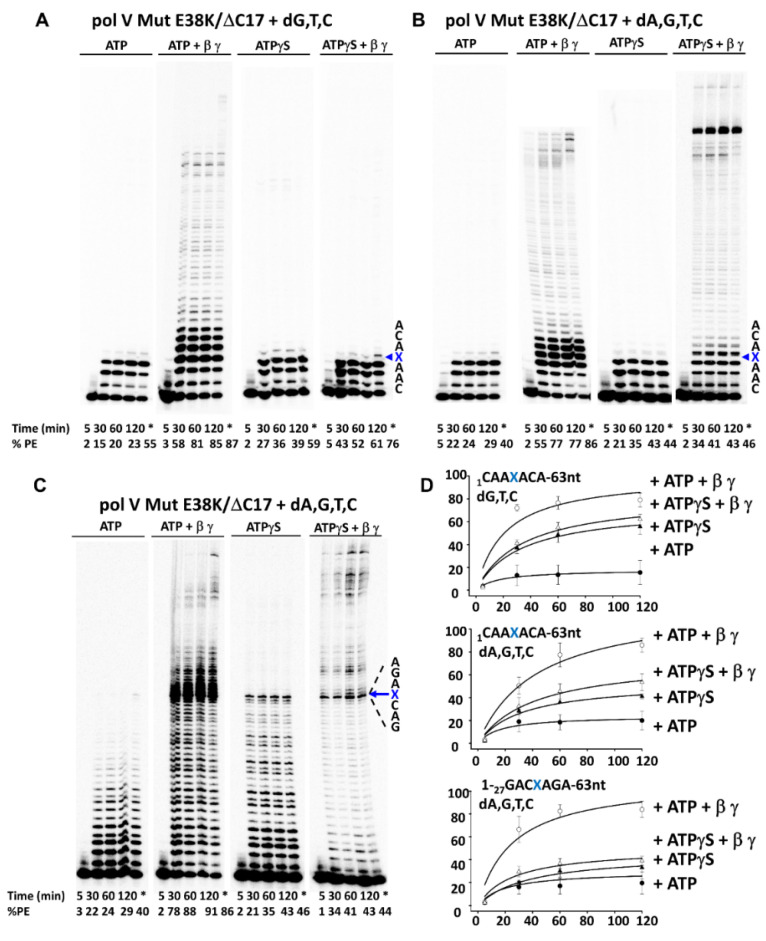
Trans-lesion synthesis (TLS) by Pol V Mut E38K/∆C17 on p/t DNA containing an abasic site. (**A**–**C**) TLS of pol V Mut WT (400 nM) was measured on circular p/t DNA (20 nt primer/ 90 nt circle; 25 nM) in the presence and absence of β-clamp and γ-complex with saturating concentration of ATP/ATPγS (500 µM) and with two sets of dNTPs: (**A**) dGTP, dTTP, dCTP and (**B**,**C**) dATP dTTP, dCTP, dGTP (500 µM each). (**A**,**B**) Activity of pol V Mut WT on p/t containing an abasic site 3 nucleotides away from 3’-OH of primer. (**C**) activity of pol V Mut WT on p/t containing abasic site 30 nucleotides away from 3’-OH of primer. The abasic site is marked by an X. Samples were taken at different times to measure DNA synthesis and at 120 min 400 nM of RecA * was added to the reaction to *trans*-activate the mutasome which is represented by *. The experiments were repeated 3 times and the average % PE (percent p/t DNA extended) with the SD (standard deviation) for each reaction time point is graphed in panel (**D**).

**Table 1 cells-10-01083-t001:** List of DNA oligomers used in the experiments.

Name	Sequence
20 mer primer	5’—TTT GGA TGA AGG TGA TTT CT—3’
Regular template	5’—ATG ACA AGA CAA GAC AAG ACA AGA CAA GAC AAG ACA AGA CAA GAC AAG AAA TCA CCA TCA TCC AAA TCC ACT AAA CCA TAT CCA TCC TCG—3’
Template V1—Abasic site 3 nt away from primer terminus	5’—ATG ACA AGA CAA GAC XAG ACA AGA CAA GAC AAG ACA AGA CAA GAC AAG AAA TCA CCT TCA TCC AAA TCC ACT AAA CCA TAT CCA TCC TC—3’
Template V2—Abasic site 30 nt away from primer terminus	5’—ATG ACA AGA CAA GAC AAG ACA AGA CAA GAC AAG ACA AGA CAA XAC AAG AAA TCA CCT TCA TCC AAA TCC ACT AAA CCA TAT CCA TCC TCG—3’
20 mer linker for circularization	5’—GTC TTG TCA TCG AGG ATG GA—3’

* X represents internal dSpacer modification by IDT which constitutes to an abasic site, tetrahydrofuran (THF), or apurinic/apyrimidinic (AP) site.

## Data Availability

The data presented in this study are available on request from the corresponding author.

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
