# Peer review of "The SOS Error-Prone DNA Polymerase V Mutasome and β-Sliding Clamp Acting in Concert on Undamaged DNA and during Translesion Synthesis"

_cells, 2021, doi:10.3390/cells10051083_

Round 1

Reviewer 1 Report

The manuscript, “The SOS Error-Prone DNA Polymerase V Mutasome and beta-sliding Clamp Acting in Concert on Undamaged DNA and During Translesion Synthesis” by Sikand et al is a fitting tribute to the foundational scientists who discovered and pioneered the E. coli SOS pathway.

The Introduction gives a wonderful historical perspective of the key experiments and subsequent thought processes that drove the discovery of the SOS pathway. This Introduction is valuable in not only establishing the foundational underpinnings of the field, but also (and even more importantly) presents a perspective of the value of collegial interactions that refine hypotheses.

This investigation, using two forms of the mutasome (pol V Mut bound to wild type RecA and pol V Mut bound to a SOS constitutive RecA mutant) examined how DNA synthesis, including TLS, was influenced by the beta-sliding processivity clamp. All studies are clearly and concisely presented and provide addition insights into the complexity of the mechanisms modulating survival at the cost of mutagenesis. With the minor recommendations below, this is an outstanding contribution from an all-star cast of current investigators in the field.

Address the following:

Line 222 needs correcting – currently appears as: 3.1.(. stimulation of pol V Mut on undamaged DNA

The reporting of the quantitation of %PE in portions of the data shown in Figure 4 Panels B, C, and E do not appear to correspond well to the gel image. In Panel B, lanes 9 and 10 the %PE are given as 10 and 52%, but the differences appear to be much less than that reported. In Panel C, lanes 17-20 appear very similar, but the %PE values are given as: 24, 35, 35, and 66%. For Panel E, it lists lane 10 as 26%, yet it appears to have the greatest extension shown in the gel image. If one compares the intensity in lanes 19 and 20 with their reported values of 26 and 61%, respectively, the value given for lane 10 seems inaccurate. The differences between these data and images should clarified. Similar concerns were NOT noted in the data presented in Figure 5.

E. coli is not uniformly italicized in the manuscript.

Author Response

We thank the reviewer for their positive comments and for their careful analysis of our data, which has allowed us to correct several gel band calculation errors. We have addressed each point raised by the reviewer.

Reviewer Comment 1: Line 222 needs correcting – currently appears as: 3.1.(. stimulation of pol V Mut on undamaged DNA

Response 1: Corrected on Page 7 line 221 and the sentence reads as follow: 3.1. ß stimulation of pol V Mut on undamaged DNA.

Reviewer Comment 2: The reporting of the quantitation of %PE in portions of the data shown in Figure 4 Panels B, C, and E do not appear to correspond dwell to the gel image. In Panel B, lanes 9 and 10 the %PE are given as 10 and 52%, but the differences appear to be much less than that reported. In Panel C, lanes 17-20 appear very similar, but the %PE values are given as: 24, 35, 35, and 66%. For Panel E, it lists lane 10 as 26%, yet it appears to have the greatest extension shown in the gel image. If one compares the intensity in lanes 19 and 20 with their reported values of 26 and 61%, respectively, the value given for lane 10 seems inaccurate. The differences between these data and images should clarified. Similar concerns were NOT noted in the data presented in Figure 5.

Response 2: We thank the reviewer for their careful analysis of our data and pointing out the inconsistencies. We revisited our raw data and recalculated the gel lanes mentioned by reviewer (please see detailed response below). All gel-based data presented in this article were analyzed as follow: the gel band intensities were measured using Phosphorimager and analyzed by ImageQuant software and the % of primer extension (% PE) was calculated by integrating the band intensities of extended primer (bands above the bottom band) divided by the total integrated band intensity (extended bands + bottom (unextended) band).

­­In Figure 4 Panel B, lane 10 was originally reported as 52% and after data reanalysis was corrected to 47%.

In Figure 4 Panel C, lane 20 was originally reported as 66% and after data reanalysis was corrected to 42%.

In Figure 4 Panel E, lane 10 was originally reported as 26% and after data reanalysis was corrected to 74%.

Reviewer Comment 3: E. coli is not uniformly italicized in the manuscript.

Response 3: Corrected in line 273 and line 532.

Reviewer 2 Report

This paper probes the role of the beta sliding clamp in polymerization and lesion bypass activity of pol V Mut (UmuD’2C-RecA-ATP). The beta sliding clamp is shown to have a stimulatory effect on the activity of pol V Mut. The authors suggest that this is because the beta sliding clamp can tether pol V to the end of the primer and because beta can inhibit pol V Mut’s ATPase activity, which is coupled to dissociation of pol V Mut from DNA. The major issue here is that there is not convincing evidence presented that beta clamp has an effect on pol V Mut ATPase activity. This seems like it could be tested directly. The group of Robert Fuchs has worked to elucidate the stimulation of pol V by beta sliding clamp, and at least some of that work should be discussed and referenced here. Some examples: Fujii 2004 EMBO J; Fujii 2004 J Mol Biol; Fujii 2009 PNAS. Other minor items to address: Page 5, lines 145 and 162-163, in both cases, “fraction” should be “fractions” Page 5, line 184, P32 (phosphorus-32) should be written in the standard way (in isotope notation). Page 7, line 222, there is a stray character in the title line. Figure 4, part C, heading/label is missing Page 14, line 433, there are extra periods at the end of the line.

Author Response

Reviewer Comment 1: This paper probes the role of the beta sliding clamp in polymerization and lesion bypass activity of pol V Mut (UmuD’2C-RecA-ATP). The beta sliding clamp is shown to have a stimulatory effect on the activity of pol V Mut. The authors suggest that this is because the beta sliding clamp can tether pol V to the end of the primer and because beta can inhibit pol V Mut’s ATPase activity, which is coupled to dissociation of pol V Mut from DNA. The major issue here is that there is not convincing evidence presented that beta clamp has an effect on pol V Mut ATPase activity. This seems like it could be tested directly.

Response 1: We thank reviewer for this comment and in the Discussion section of the article, on page 14, lanes 398-404, we have added two sentences addressing challenges of creating an experimental set up to measure pol V Mut ATPase rate in the presence of ß-clamp. We wrote:­­

“To address the question experimentally whether ß affects the rate of DNA-dependent ATPase of pol V Mut, a new experimental set-up needs to be designed where we can distinguish between the weak rate of pol V Mut ATPase from the much stronger ATPase activity of γ-complex used to load ß on DNA. This promises to be a challenging task but one which is certainly important to address in the future to determine if and how the intrinsic DNA-dependent ATPase of pol V Mut modulates polymerase activity.”

Reviewer Comment: The group of Robert Fuchs has worked to elucidate the stimulation of pol V by beta sliding clamp, and at least some of that work should be discussed and referenced here. Some examples: Fujii 2004 EMBO J; Fujii 2004 J Mol Biol; Fujii 2009 PNAS.

Response 2: On page 13 we added two sentences along with appropriate references mentioning ours’ and Robert Fuchs’ studies of the stimulatory effect of beta clamp on the activity of DNA polymerase V in the presence of a RecA* nucleoprotein filament. In this work however, we have shown for the first time that ß-clamp stimulates the activity of the stand-alone active form of pol V (pol V Mut) which synthesizes DNA in the absence of RecA*. On page 13, lines 343 to 346 we state:

“Previous studies have shown a stimulatory effect of b-clamp on the processivity of pol V in the presence of RecA* [36,45,47-49]. Here, we report for the first time a strong stimulatory effect of ß-clamp during synthesis on undamaged DNA and TLS using the stand-alone active form of pol V, i.e., pol V Mut, in the absence of RecA*.”

Reviewer Comment 3: Other minor items to address: Page 5, lines 145 and 162-163, in both cases, “fraction” should be “fractions”.

Response 3: Corrected on Page 5 in lines 142 and line 165.

Reviewer Comment 4: Page 5, line 184, P32 (phosphorus-32) should be written in the standard way (in isotope notation).

Response 4: Corrected on Page 5, line 183

Reviewer Comment 5: Page 7, line 222, there is a stray character in the title line.

Response 5: Corrected on Page 7 line 221 and the sentence reads as follow: 3.1. ß stimulation of pol V Mut on undamaged DNA

Reviewer Comment 6: Figure 4, part C, heading/label is missing.

Response 6: Corrected and replaced with new Figure.

Reviewer Comment 7: Page 14, line 433, there are extra periods at the end of the line.

Response 7: Corrected on Page 14, line 441.

Round 2

Reviewer 2 Report

The authors have revised the manuscript, which is a lovely tribute to Miroslav Radman.

One other revision should be made, which is to clearly delineate that the two roles highlighted here are supported by evidence in the first case and speculation in the second:

The results suggest that b may have two stabilizing roles: its canonical role in tethering the pol at a primer-3'-terminus, and a second role in inhibiting pol V Mut’s ATPase to reduce the rate of mutasome-DNA dissociation.

Two possible revisions:

The results suggest that b may have two stabilizing roles: its canonical role in tethering the pol at a primer-3'-terminus, and possibly [or: we propose] a second role in inhibiting pol V Mut’s ATPase to reduce the rate of mutasome-DNA dissociation.

Author Response

Reviewer Comments: One other revision should be made, which is to clearly delineate that the two roles highlighted here are supported by evidence in the first case and speculation in the second:

The results suggest that b may have two stabilizing roles: its canonical role in tethering the pol at a primer-3'-terminus, and a second role in inhibiting pol V Mut’s ATPase to reduce the rate of mutasome-DNA dissociation.

Two possible revisions:

The results suggest that b may have two stabilizing roles: its canonical role in tethering the pol at a primer-3'-terminus, and possibly [or: we propose] a second role in inhibiting pol V Mut’s ATPase to reduce the rate of mutasome-DNA dissociation.

Response: We thank the reviewer for their comments and have made the appropriate changes, highlighted in red in the manuscript. Specifically, in the abstract we have added “a possible” (Page 1, line 28), and in the Discussion we have added “This stabilizing effect of ß suggests perhaps that along with its canonical role in tethering pol V Mut at 3'-primer termini, ß might have a possible second role in inhibiting the intrinsic DNA-dependent ATPase of pol V Mut to reduce the rate of mutasome dissociation” (Page 14, lines 443-445).
